# Uveal Melanoma Metastasis

**DOI:** 10.3390/cancers13225684

**Published:** 2021-11-13

**Authors:** Ernesto Rossi, Michela Croce, Francesco Reggiani, Giovanni Schinzari, Marianna Ambrosio, Rosaria Gangemi, Giampaolo Tortora, Ulrich Pfeffer, Adriana Amaro

**Affiliations:** 1Medical Oncology, Fondazione Policlinico Universitario Agostino Gemelli IRCCS, 00168 Rome, Italy; ernesto.rossi@policlinicogemelli.it (E.R.); giovanni.schinzari@policlinicogemelli.it (G.S.); giampaolo.tortora@policlinicogemelli.it (G.T.); 2Laboratory of Biotherapies, IRCCS Ospedale Policlinico San Martino, 16132 Genova, Italy; michela.croce@hsanmartino.it (M.C.); rosaria.gangemi@hsanmartino.it (R.G.); 3Laboratory of Epigenetics, IRCCS Ospedale Policlinico San Martino, 16132 Genova, Italy; Francesco.reggiani@hsanmartino.it (F.R.); marianna.ambrosio.94@outlook.it (M.A.); adriana.amaro@hsanmartino.it (A.A.); 4Medical Oncology, Università Cattolica del S. Cuore, 00168 Rome, Italy

**Keywords:** uveal melanoma, tumorigenesis, molecular classification, targeted therapy, immune checkpoint

## Abstract

**Simple Summary:**

Survival after diagnosis of metastatic uveal melanoma has not significantly improved over decades, most patients die within a year from diagnosis. Uveal melanoma is clearly distinct from cutaneous melanoma and the therapies developed for the latter do not work for the former. This is apparently due to three major aspects of UM: (i) the mutations that drive UM tumorigenesis activate two oncogenic signaling pathways and one of the two cannot yet be targeted by therapy; (ii) UM has relatively few tumor specific neo-antigens that could be presented to the immune system and therefore evades immune control; and (iii) UM shows an infiltrate of inflammatory cells that promote tumor progression. However, in the future, there might be drugs to target both oncogenic pathways that are activated in UM, new immune therapy approaches might circumvent the paucity of neo-antigens and liver directed, local therapy might improve response to cytostatic therapy. Hopefully, there will be therapies that can repeat the success obtained with targeted and immune therapy for cutaneous melanoma.

**Abstract:**

Uveal melanoma (UM) is characterized by relatively few, highly incident molecular alterations and their association with metastatic risk is deeply understood. Nevertheless, this knowledge has so far not led to innovative therapies for the successful treatment of UM metastases or for adjuvant therapy, leaving survival after diagnosis of metastatic UM almost unaltered in decades. The driver mutations of UM, mainly in the G-protein genes GNAQ and GNA11, activate the MAP-kinase pathway as well as the YAP/TAZ pathway. At present, there are no drugs that target the latter and this likely explains the failure of mitogen activated kinase kinase inhibitors. Immune checkpoint blockers, despite the game changing effect in cutaneous melanoma (CM), show only limited effects in UM probably because of the low mutational burden of 0.5 per megabase and the unavailability of antibodies targeting the main immune checkpoint active in UM. The highly pro-tumorigenic microenvironment of UM also contributes to therapy resistance. However, T-cell redirection by a soluble T-cell receptor that is fused to an anti-CD3 single-chain variable fragment, local, liver specific therapy, new immune checkpoint blockers, and YAP/TAZ specific drugs give new hope to repeating the success of innovative therapy obtained for CM.

## 1. Introduction

Uveal melanoma accounts for approximately 5% of all human melanomas. It is the most frequent non-cutaneous site of melanoma development (for recent reviews see [1,2]). The incidence in the USA is 0.43 per 100,000 [3]. In Europe, incidence follows a North–South gradient from 0.8 to 0.2 per 100,000 [4] likely due to the higher prevalence of light skin color and lightly pigmented irises in northern-European countries. Risk factors are light skin and eye color [5] and professional activity as a welder [6,7]. A genome wide association study identified loci linked to these skin and eye pigmentation [8]. Exposure to sun light during work or leisure time is not significantly linked to UM risk [5].

UM develops from uveal melanocytes that derive from the neural crest during embryonal development just like the melanocytes from which CM originates [9,10]. UM and CM share the risk factors linked to skin color yet the etiological function of UV-radiation, well established for CM [11], is unlikely to contribute to the development of UM [7]. The lens and the vitreous body almost completely absorb UV wave lengths [12]. Absorption may decrease in elder persons. UV etiology can also be excluded for UM by considering the mutational signatures: CM and UM show prevalently C > T transition mutations yet in different sequence contexts [13,14]. Hence, UM does not show the mutational signature that is linked to UV-exposure [14].

Therapy of primary UM is associated with an excellent control of the disease and local recurrence is very infrequent. Dissemination of cells from primary UM occurs via the bloodstream not via lymphatic vessels. Approximately 50% of patients with primary UM die within 10 years after diagnosis from UM metastases. UM–related mortality was 31%, 45%, and 49% 5, 15, and 25 years, respectively, according to cumulative incidence analysis [15].

UM shows a very limited number of molecular lesions that drive progression to metastasis (see below). These alterations define molecular classes that are associated with metastatic risk and their analysis allows for a precise prognostication [16]. Nonetheless, development of UM therapy has not kept the path of the innovation in prognostication and survival after diagnosis of metastatic UM has not dramatically changed in decades [17]. Until recently, there were no FDA approved therapies for metastatic UM [18,19]. This has changed with the recent acceptance of a biologic license application for Tebentafusp in metastatic UM by the FDA [20]. Tebentafusp, a chimeric protein that binds to the melanocytic protein gp100 expressed by most melanoma cells, and to the T-cell receptor thereby mediating the recognition of the tumor cells by T-cells, has shown promising activity in clinical trials for UM [21,22]. The drug is restricted to patients carrying a specific human leukocyte antigen (HLA)-A type (*0201) [21]. Still, there are no approved adjuvant therapies.

Targeted therapies and immunotherapies that dramatically improved survival for patients affected by CM showed only limited effects for the therapy of UM [18,23,24]. Immune checkpoint blockers might be efficacious in UM with exceptionally high mutational burden, such as UM carrying mutations in the Methyl-CpG Binding Domain 4 (MBD4) gene [25,26]. New therapies targeting immune checkpoints that are more relevant for UM, such as LAG3, may be more promising [27,28].

## 2. Metastasis Associated Molecular Characteristics of UM

UM shows molecular features that are strikingly different from CM and these differences are believed to determine the drastically different responses to targeted and immune therapy. The mutations that likely initiate UM tumorigenesis occur in a mutually exclusive manner in two genes encoding alpha-subunits of G-proteins, GNAQ [29] and GNA11 [30], or much less frequently in the Cysteinyl Leukotriene Receptor 2 (CYSLTR2) [31] or the Phospholipase C Beta 4 (PLCB4) [32] genes. All four initiator genes act in the same pathway of G-protein coupled receptor signaling. The downstream effectors are the mitogen protein kinase (MAP-kinase) pathway [33] and the YAP/TAZ pathway of organ size control, the latter in a HIPPO independent manner [34,35]. CM shows different initiator mutations, mainly the B-Raf Proto-Oncogene (BRAF), NRAS proto-oncogene GTPase (NRAS), and neurofibromin 1 (NF1) genes that also signal via MAP-kinases [36]. Signaling via YAP/TAZ has not been reported for CM. GNAQ and GNA11 mutations are also found in blue nevi [37], in a rare subtype of CM [38] and in melanocytic tumors of the central nervous system [39] but not in conjunctival melanoma [40] that resembles CM. Mutations in the promoter of the Telomerase Reverse Transcriptase (TERT) gene are common in CM and very rare in UM [41]. The initiator mutations of CM and UM do not necessarily reflect the prevalent mutation signature as they are generated by random DNA replication errors rather than as a consequence of the etiological agent, ultraviolet light in CM, unknown for UM. Apparently, not only in melanoma, the tissue context determines which randomly generated mutations can start tumorigenesis.

UM, in contrast to CM, shows a very low mutational burden of 0.5 mutations per megabase [32]. The initiator mutations cited appear to be sufficient for tumor development. For the acquisition of the metastatic phenotype a single further mutation, either in the BRCA1 associated protein 1 (BAP1) [42] or in the splicing factor 3B1 (SF3B1) [43,44] genes and DNA copy number alterations convey a greatly increased risk of metastasis. BAP1 mutations are associated with UM and CM in an opposed manner, with poor outcome in UM and better outcome in CM [45]. A few other rare mutations [46], many of which occur in genes encoding members of the calcium signaling pathway [14], may also contribute to UM tumorigenesis. BAP1 confers a high risk of metastasis to UM whereas SF3B1 determines an intermediate risk of metastasis that generally occurs with a longer latency [47]. The comparison of primary with metastatic UM by whole genome sequencing has not evidenced frequent additional mutations in metastatic lesions [46] in accordance with a similar study that used targeted re-sequencing [48]. Metastases may, however, show additional mutations or copy number alterations (among which, interestingly, a further amplification of the risk associated chr8q). Dissemination of cells starts early during the evolution of the primary tumor [49,50], leading to a complex tumor heterogeneity despite the paucity of somatic mutations [51]. Mutational heterogeneity in multiple UM metastases, however, hardly compares to the big bang model, also referred to as punctuate equilibrium evolution, where several heterogeneous clones outgrow after a generalized genetic instability [52,53].

In addition to somatic mutations, a few, very characteristic DNA copy number alterations have a high impact on metastatic progression of UM. Monosomy of chromosome 3 is a paramount feature of UM of high risk of metastasis [54]. BAP1 is located on chr3 so that most metastatic UM carry a single, mutated copy of the gene. Amplification of the long arm of chromosome 8 (chr8q) is also strongly associated with metastatic risk [55], both in the absence and in the presence of chr3 monosomy [56]. Increased risk of UM with Chr8q gain is determined at least in part by overexpression of the ArfGAP With SH3 Domain, Ankyrin Repeat And PH Domain 1 (ASAP1, also termed DDEF1) gene [57]. The oncogene MYC that is located in 8q24.21 is not over-expressed in UM with chr8q gain [58]. Chr6p gain is associated with a more benign behavior of UM with chr3 monosomy [59]. Chr6p gain and chr3 monosomy have been reported to be almost mutually exclusive [60] yet the analysis of two datasets that contain these data [16,41] shows frequent concomitance of the two chromosomal alterations (Figure 1). Chr6p harbors the cluster of HLA genes [61,62] as well as immune regulatory genes of the B7 family [63]. HLA expression is not consistently altered by chr6p gain and, in contrast to the general rule, HLA downregulation and allele loss are not associated with worse outcome, hinting at prevalent tumor control by natural killer cells [64]. Inflammation of the primary tumor, that initially grows in an immune-privileged site [65] (see below), appears to be mainly pro-tumoral and is associated with a higher risk of metastasis [66,67]. Inflammation defines a subgroup of high-risk UM that is also characterized by chr8q gain [61] and that constitutes a distinct molecular subgroup within the high-risk class, detectable by gene expression profiling and methylome analysis [16]. Figure 1 summarizes the co-occurrence of the five most important molecular features of primary UM with high risk of metastases, BAP1 and SF3B1 mutations and chr3, 6, and 8 copy number alterations. This analysis shows that all features occur in concomitance in at least one of 48 primary tumors that have developed metastases so that no absolute exclusivity can be claimed for none of these molecular lesions (Figure 1).

The co-occurrence of monosomy of chromosome 3 (M3), chromosome 6p amplification (Chr6p), chromosome 8q amplification (Chr8q), BAP1, and SF3B1 mutations was analyzed for 133 cases (left panel), the 84 of which without development of metastases during follow-up (center panel) and the 48 cases with metastases (right panel) using VENN-diagrams (http://bioinformatics.psb.ugent.be/webtools/Venn/ accessed on 5 October 2021). A high grade of association of M3, BAP1, and chr8q is evident in cases with metastases. No case with only chr6p gain developed metastases yet all cases with chr6p gain that developed metastases also showed at least one other feature of metastatic cases.

High-risk UM also shows a distinct DNA-methylation profile that is reflected by transcriptional changes [16,68]. BAP1 mutations are associated with a different DNA-methylation pattern. BAP1 itself is regulated by DNA-methylation [69] and its knock-down in vitro introduces methylation alterations similar to those observed in high risk UM [70]. Histone deacetylase (HDAC) inhibitors can apparently at least partially revert the effect of the BAP1 mutation in terms of chromatin structure [71] leading, among others, to the induction of major histocompatibility complex class 1 (HLA-1) genes [72]. The dependency of BAP1 mutated UM on the Enhancer Of Zeste 2 Polycomb Repressive Complex 2 Subunit (EZH2) and hence the possibility to treat UM with EZH2 inhibitors has been controversially discussed [73,74]. Recent evidence obtained by single cell transcriptomics invokes the Polycomb Repressive Complex 1 (PRC1) and the associated ubiquitination state of lysine residue 119 in histone H2a as a major driver of UM progression [75]. Histone H2a is ubiquitinated by PCR1 and deubiquitinated by BAP1 [75]. A more complex genomic substructure of UM also emerges from another study based on single cell transcriptomics [28].

The combined analysis of copy number, transcriptomic, and methylomic data using data fusion approaches does not improve molecular classification of UM [68]. Each single domain is likely sufficient for prognostic classification. In fact, the status of chr3 and 8 is routinely analyzed by multiplex ligation-dependent probe amplification (MLPA) [76] and, in alternative, gene expression data of 15 genes can be used for prognostication in the clinical setting [77]. The dimension of the tumor, especially the basal diameter, remains, however, an independent prognostic factor [78].

## 3. Pathophysiology of UM Metastasis

Most UM metastases are estimated to be initiated up to five years before diagnosis or treatment of the primary tumor [79,80] and indeed, circulating tumor cells can be found in UM patients before clinical signs of metastatic disease [81,82]. UM cells that leave the eye reach the liver and may remain quiescent for years. Dormancy can be considered a mechanism of adaptation to stress used by tumor cells to survive in a hostile microenvironment. Dormant UM cells regulate proliferation, activate autophagy, and contrast hypoxia by inducing angiogenesis. During the latency, disseminated tumor cells escape immunity. Changes in the microenvironment, chronic inflammation, and immune evasion can eventually activate the growth of locally adapted micro metastases [83,84].

Liver is the most frequent metastasis target tissue [85]. This tropism might depend on the expression of certain receptors (such as MET Proto-Oncogene (c-Met), Insulin Like Growth Factor 1 Receptor (IGF-1R), C-X-C Motif Chemokine Receptor 4 (CXCR4)) whose corresponding ligands are expressed in the liver (Hepatocyte Growth Factor (HGF), Insulin Like Growth Factor 1 (IGF1), C-X-C Motif Chemokine Ligand 12 (CXCL12)). Recent data have also shown that exosomes from UM cells expressing integrin aVb5 enter into the liver to establish a pre-metastatic niche promoting liver tropism [86,87].

The escape of UM cells from immune surveillance relies on alteration of the major histocompatibility (MHC) class I antigen expression on UM cells. Cells with low MHC are highly susceptible to elimination by natural killer (NK) cells, but in the eye, they are protected by the presence of the Macrophage Migration Inhibitory Factor (MIF) and the Transforming Growth Factor Beta 1 (TGFβ1) in the aqueous humor that negatively regulates NK cell functions. High concentrations of NK cells in the liver select for UM cells with high MHC class I expression that become the dominant phenotype [88].

Both intrinsic characteristics of UM metastases and their interactions with host cells have been studied to understand why the liver environment is so attractive for UM cells. Grossniklaus and colleagues distinguished infiltrative or nodular growth patterns when analyzing UM liver metastases of 15 patients. In the infiltrative pattern, UM cells lack vascular endothelial growth factor (VEGF) expression. They invade the perisinusoidal space in the liver to get access to nutrients and oxygen and create pseudo-sinusoidal spaces. In the nodular pattern, UM metastases originate in the peri-portal area, increase the expression of Matrix Metallopeptidase 9 (MMP9) and VEGF thus acquiring angiogenic properties [89]. Hepatic stellate cells also contribute to the UM niche in the liver by secreting pro-inflammatory factors and collagen and by stimulating angiogenesis [90].

The Melanoma Associated Antigen Gene (MAGE) family proteins, tyrosinase, and the Premelanosome Protein gp100 are UM tumor-associated antigens (TAA) that are recognized by cells of the immune system [91]. Peripheral CD8+ cells from UM patients and tumor-infiltrating lymphocytes (TILs) can lyse UM cells in vitro [92,93]. Yet the immune privilege of the eye allows UM cells to escape the control by the immune system. The blood–retina barrier, the absence of efferent lymphatics [94], the presence of soluble factors (TGF-β [95,96]), low MHC expression and expression of FAS ligand [97] contribute to protect eyes from severe inflammation. Anterior chamber-associated immune deviation (ACAID) induces complex immunoregulatory activities [1,98]. Cytokines present in the aqueous humor, such as MIF, inhibit NK cell activity [99] whereas cells of the iris and of the ciliary body prevent T cell activation and proliferation through cell–cell contacts [100]. The immune-suppressive microenvironment of the eye forms a niche in which UM grows despite innate and adaptive immunity and disseminates after breaking the blood–retina barrier. Expression of indoleamine dioxygenase-1 (IDO-1) [101,102] and PD-L1 [103] enhance the metastatic potential of UM cells that leave the eye.

The immunomodulatory nature of the liver is determined by its exposure to food antigens, allergens, and high levels of gut derived endotoxins. The liver microenvironment is composed of resident non-immune and immune cells—such as hepatocytes, liver sinusoidal endothelial cells (LSECs), Kupffer cells (KCs), T, NK, and natural killer T (NKT) cells—that strictly regulate the balance between tolerance and the defense against pathogens. UM cells that have escaped from the eye apparently find further protection in the immune-modulatory microenvironment of the liver. Metastatic cells can trigger liver specific tolerance mechanisms to suppress systemic anti-tumor T cell immunity. This may explain why cancer patients with liver metastases have a substantially worse prognosis even when treated with anti-PD-1 immunotherapy [104].

The dogma that links tumor infiltrating lymphocytes (TILs) to a better prognosis does not apply to UM. Tumor infiltrating lymphocytes (TIL) are regulatory rather than cytotoxic [105] or “exhausted” [28,89,106,107] CD8+ T lymphocytes. Pro-tumoral M2 macrophages are prevalent in the tumor microenvironment (TME) of UM whereas CD4+ and CD8+ lymphocytes are restricted to the periphery of the tumor and where they cannot attack tumor cells [108]. However, a high metastatic infiltration of CD8+ T cells and macrophages (CD68+) correlated with longer overall survival for patients undergoing isolated hepatic perfusion as treatment of liver metastases [109,110]. Yet most transcriptomic and bioinformatic approaches have shown a correlation between TILs and short survival [111]. A different scenario appeared when studying the functional properties of T-cells and macrophages in depth and measuring tumor-immune cell distances in liver and extra liver metastases. A multiplex fluorescence immunohistochemistry (mIHC) analysis of 21 metastatic UM found a similar fraction of activated cytotoxic T cells (CD8 + Granzyme B+ CTL) in the intra-tumoral area of metastatic UM patients with progressive or stable disease but a higher intratumoral versus peritumoral ratio of the same T cells in patients with stable disease. In the same study, patients with lower numbers of intratumoral CD4+ T-regulatory cells (CD4+ T_reg_) survived for longer times when treated with immunotherapy [112]. The percentage and localization of activated cytotoxic and regulatory T cells might be predictive of the response to immunotherapy. In an immunocompetent mouse model, the anti-PD-1 driven anti-tumor responses were inhibited by the presence of T_regs_ and myeloid derived suppressor cells (MDSC). These results sustain the necessity to combine anti-PD-1 therapy with agents targeting T_reg_ and MDSC [104]. At the same time, they call for a deeper and more accurate understanding of liver TME in order to identify new targets for future combination therapies.

## 4. Therapy of UM Metastases

About 50% of patients with UM develop metastases [15]. The liver is usually the first and frequently the only metastatic site [15]. Patients often die of hepatic failure [113]. A recent single institution’s longitudinal analysis of UM patients with liver metastases shows a slight improvement in therapy outcome likely due to the introduction of liver directed therapies instead of dacarbazine based chemotherapy [114,115]. Several of these therapies have been evaluated, mainly in retrospective, uncontrolled studies and in various clinical settings [115]. In particular, surgery alone or in combination with another local treatment—such as transarterial embolization, selective internal radiotherapy, isolated hepatic perfusion, hepatic artery infusion, and immune-embolization—has been employed with benefit in terms of prolonged survival only in selected patients [115]. In a retrospective analysis, liver resection yielded an overall survival of 14 months, 27 months when R0 resection was obtained [116]. In the studies evaluating transarterial embolization, median survival ranged from 5 to 29 months [115,117,118]. Selective internal radiotherapy yielded a median survival from 9 months to 24 months [115,119,120]. Isolated or percutaneous perfusion and hepatic artery infusion (with fotemustine, cisplatin+vinblastine+dacarbazine, carboplatin, melphalan, or other combinations) has been employed with a median survival from 9 to 25 months [115,121,122]. Immuno-embolization allowed a median survival of up to 21 months in the study conducted by Valsecchi [123].

Unfortunately, not all patients can undergo locoregional treatments (i.e., patients with multiple metastatic sites). Therefore, many attempts with systemic therapy have been performed. Poor results have been obtained with chemotherapy. The median survival with first-line fotemustine was 13.9 months [124] (see also Table 1). A three-drug first-line chemotherapy allowed an overall survival (OS) of 13 months in the whole population and 21 months for patients achieving an objective response [113].

MET inhibitors have been tested given the high expression of the receptor kinase on UM cells [125]. Cabozantinib has been compared with chemotherapy, without advantage in terms of progression-free survival (PFS) [126]. Other targeted agents did not demonstrate a relevant clinical improvement. Selumetinib, a mitogen activated kinase kinase (MEK) 1–2 inhibitor, determined a slight benefit in PFS versus chemotherapy, without significant difference in OS (11.8 vs. 9.1 months) [127]. In a study including mostly pretreated patients, OS and PSF with sunitinib were 8.2 and 4.2 months, respectively [128]. Combination of drugs, such as chemotherapy and targeted agents, did not yield a remarkable improvement in survival. The survival obtained by the association of bevacizumab and temozolomide was 10 months [129], the combination of selumetinib and dacarbazine did not add an advantage in terms of PFS and OS compared to dacarbazine alone [130]. The association of chemotherapy (bleomycin, vincristine, lomustine, dacarbazine) with human leukocyte interferon demonstrated modest activity, with 12 months of OS [131] (see also Table 1). 

Based on the effectiveness in cutaneous melanoma, immune checkpoint inhibitors are widely used also for UM [132]. Ipilimumab (anti-CTLA4-antibodies) in pretreated patients showed an OS of 6 months, with 1-year survival rate of 31% [133]. The anti-PD-1 agent pembrolizumab was evaluated in a prospective observational study, obtaining a median PFS of 3.8 months, while OS for patients with clinical benefit was 12.8 months [134]. The results reported for the combination of ipilimumab and nivolumab (anti-PD1-antibodies) were 12 months of OS and 3.0 months of PFS [135] (see also Table 1).

A series of 14 metastatic UMV patients was treated with dendritic cell vaccination, using antigen-presenting cells loaded with gp100 and tyrosinase: a tumor specific response was observed in 29% of the patients, while median overall survival was 19.2 months [86]. A phase I trial with dendritic cell vaccination, in addition to checkpoint inhibitors or chemotherapy, is ongoing (NCT04335890).

Recently, the T-cell directed therapy (Tebentafusp, a T-cell receptor fused to an anti-CD3 effector) showed promising results, with a 1-year survival rate of 73% versus 59 for the control arm (dacarbazine or pembrolizumab or ipilimumab) in a phase III study [22]. A biologic license application for Tebentafusp in metastatic UM has been accepted by the FDA [20]. The oncolytic adenovirus ICOVIR5, which may be able to exert a lytic activity through replication within tumor cells and promote immune response, has been employed in a Phase I study. It showed the ability to reach the metastatic sites, yet tumor regression was not observed [136]. Other oncolytic viruses are under investigation in clinical studies [138] (see also Table 1).

Current trials are also exploring the role of further targeted agents, such as the protein kinase C inhibitor IDE196 in patients with GNAQ/GNA11 mutations or protein kinase C fusions, PARP inhibitors alone or in combination with Nivolumab, PLX2853, an inhibitor of bromodomain-containing protein-4 (BRD4), an epigenetic regulator belonging to BET family [138]. The combination of Pembrolizumab with the histone deacetylase inhibitor Etinostat resulted in durable responses in a subset of patients with metastatic UM (objective response rate of 14%). In the whole population of the study, PFS was 2.1 months and OS 13.4 months [137]. Ongoing clinical trials are also testing different strategies of immunotherapy and treatments including liver directed therapies [138,139].

Systemic therapy for metastatic UM is clearly unsatisfying. None of the treatments discussed above can yet be considered standard. A deeper knowledge of the metastatic disease is warranted to obtain better results with tailored systemic therapies.

## 5. UM Metastasis Models

The use of spheroids, in vitro 3D culture systems, to simulate the in vivo tumor growth is a useful method to study the effects of drugs before they come to the clinic [140]. 3D co-culture approaches showed that UM cell lines can migrate along vascular tubules generated in vitro by human endothelial cells (HUVEC), co-cultured with UM cells [141]. UM cell lines grown as spheroids and embedded into collagen were used to test different types of treatments [142,143]. Three-D cultures, however, do not resemble the complex multicellular microenvironment of UM metastases and should include stromal, hepatic stellate, and endothelial cells to better model metastatic lesions.

Animal models of UM are needed to understand the biological mechanisms of the metastatic process and to test new therapeutic approaches. Mouse models can be syngeneic, xenografted, genetically engineered, or humanized. Syngeneic mouse models, where murine tumor cells and mice share the same genetic background, allow for the interplay between the tumor and the immune system. Existing UM syngeneic murine models are obtained using murine cutaneous melanoma cell lines to generate liver metastasis, yet they resemble neither the biology nor the mutational landscape of UM [144]. Moreover, these syngeneic models do not even recapitulate the interactions within the TME between tumor and immune cells, nor is it possible to study the response to immunological therapies.

More recently, genetically engineered models for UM have been developed by introducing Gna11 or Gnaq mutations in mice. Mice with melanocyte-specific expression of Gna11 Q209L with and without homozygous Bap1 loss, developed tumors in the skin, eye, leptomeninges, and at the lymph nodes and lungs [145]. Mice bearing the expression of oncogenic Gnaq Q209L under control of the Rosa26 promoter were crossed to Mitf-cre/+ animals. Tumors in the eye, leptomeningeal tumors, and metastases in the lung were observed [146]. These models allow for studies of UM formation and dissemination, but metastases identified in these mice occur preferentially in the lungs, thus still not resembling human UM [145,147,148].

Xenografts of human UM cell lines, primary cell lines, or patient derived biopsies have been developed in immunodeficient mice and are mainly used for drug testing. However, efficacious therapies in preclinical xenograft models have been disappointing in clinical trials [149].

Patient-derived xenografts (PDX) are obtained by transplanting fragments of biopsies either subcutaneously or orthotopically into highly immunodeficient mice. These models are very challenging to develop for UM because they are expensive and show low engraftment rates. They are useful to test the tumor response to targeted therapy rather than to immunotherapy, since PDX cannot maintain living human immune cells [150]. Humanized mice, immune-deficient mice in which human immune cells are inoculated, may be used to overcome this issue, with the limit of rapid development of graft versus host disease and interspecies differences.

Zebrafish models have recently been developed because of the relative ease of manipulation of the embryo and the high genetic conservation with humans [151]. Zebra-fish UM models evidenced the strong migratory potential of metastatic UM cells such as OMM 2.3 or OMM 2.5 cells [152]. Transgenic zebrafish obtained by introducing Gnaq Q209L or Gna11 Q209L oncogenic alleles into the zebrafish genome under the control of the melanocyte-specific promoter mitfa developed multiple primary tumors derived also from non-ocular melanocytes [153], without metastases to the liver. The chick embryo chorioallantoic membrane (CAM) assay was also used to study growth and invasion of UM cell lines [154]. Unfortunately, no animal model of human UM that carries the typical molecular lesions and reflects the liver tropism is available as yet.

## 6. Outlook

Activation of two different oncogenic pathways, MAP-kinase and YAP/TAZ [34,35], a low mutational burden [32], a consequent low number of neo-antigens that could be recognized by the immune system, and a pro-tumoral infiltrate [61] make metastatic UM a difficult-to-treat neoplasia [24]. Therapies targeted at the MAP-kinase pathway activated by the initial tumorigenic lesions did not show major activity in the clinics, probably due to the fact that these lesions also activate the YAP/TAZ pathway. Drugs that target YAP/TAZ have been identified in preclinical studies [155,156,157] but have not yet been tested in clinical studies. Eventually, double targeting might yield results for UM comparable to those obtained with the combination of BRAF and MEK-inhibitors for CM that induce impressive responses followed, however, in most cases by development of resistance [158]. Immune checkpoint blockers (ICB) yield only limited, yet nonetheless important, responses in UM [23]. Responses might be limited to those cases that show a higher mutational burden, such as the few cases with mutations in the MBD4 gene and might—among others—rely on new ICBs that target LAG3 [27,159]. Combination of epigenetic drugs, such as Guadecitabine with ICBs, shows interesting activity in early trials for CM [160] and might also be considered for UM. Tebentafusp is likely to play a prominent role in UM therapy in the near future [21]. Local liver specific therapy might also contribute to increased survival of patients affected by metastatic UM [115]. Most likely, several of these approaches will reach approval for the treatment of metastatic UM and must be applied in a personalized and sequential manner. Hopefully, in the next 10 years, we will see a significant increase in survival after diagnosis of metastatic UM, comparable to what has been obtained for CM with targeted and immune therapies.

## Figures and Tables

**Figure 1 cancers-13-05684-f001:**
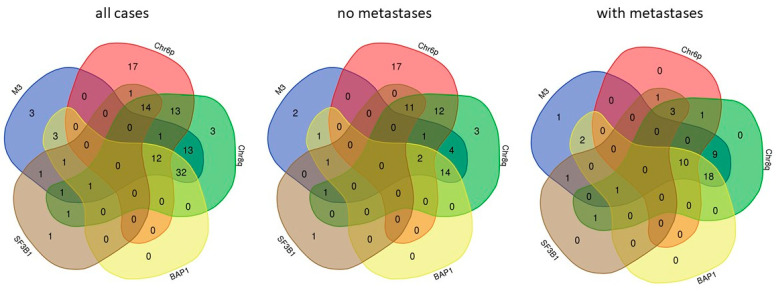
Co-occurrence of the five major molecular features of high-risk UM.

**Table 1 cancers-13-05684-t001:** Systemic Therapy for uveal melamoma.

Author	Treatment	Setting	N. pts	Objective Response (%)	PFS (m)	OS (m)
Spagnolo, F. et al. [124]	fotemustine	Naïve	25	8	-	13.9
Schinzari, G. et al. [113]	cisplatin + dacarbazine + vinblastine	Naïve	25	20	5.5	13.0
Luke, J.J. et al. [126]	cabozantinib	Pretreated	31	0	2	6.4
Carvajal, R.D. et al. [127]	selumetinib	Naïve and pretreated	69	14	4	11.8
Mahipal, A. et al. [128]	sunitinib	Naïve and pretreated	20	5	4.2	8.2
Piperno-Neumann, S. et al. [129]	bevacizumab + temozolomide	Naïve	36	0	3	10
Carvajal, R.D. et al. [130]	selumetinib + dacarbazine	Naïve	97	3	2.8	-
Pyrhönen, S. et al. [131]	bleomycin + vincristine + lomustine + dacarbazine + IFN	Naïve and pretreated	22	15	4	12
Maio, M. et al. [133]	ipilimumab	pretreated	84	5	3.6	6.0
Rossi, E. et al. [134]	pembrolizumab	Naïve	17	11.7	3.8	nr
Piulats, J.M. et al. [136]	ipilimumab + nivoluimab	Naïve	52	11.5	3	12.7
Bol, K.F. et al. [86]	dendritic cell vaccination	Naïve and pretreated	14	-	-	19.2
Nathan P. et al. [22]	tebentafusp	Naïve	252	9	3.3	21.7
Ny, L. et al. [137]	pembrolizumab + etinostat	Naïve and pretreated	29	14	2.1	13.4

pts: patients; m: months; PFS: progression-free survival; OS: overall survival; m: months; IFN: interferon; nr: not reached.

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
