# Peer review of "Uveal Melanoma Metastasis"

_cancers, 2021, doi:10.3390/cancers13225684_

Round 1
Reviewer 1 Report
The authors have done a commendable job with an extensive review of uveal melanoma metastases. It will make a valuable addition to the literature, but I believe it requires some additional work to improve its readibilty.
My global view of the manuscript is that the facts are not well linked together to tell a coherent narrative. There are also many occasions were language use is not precise enough – either using qualifiers (might, may, apparently) that don’t adequately reflect the data, or being more definitive than the evidence might support (e.g the premise about why therapeutic modalities do not work in uveal melanoma, these are certainly contributing factors but are not the whole story.
Specific, but not exhaustive, feedback follows:
INTRODUCTION
P3L59 - …that allow for a precise prognostication, unmatched by other cancers.
Provide additional support – is it truly unmatched?
P3L62 - there are FDA approved therapies neither for metastatic UM nor for adjuvant therapy
The phase III trial of tebentafusp was recently published (NEJM 9/2021) and has demonstrated an overall survival benefit. This technically has FDA accelerated approval. The manuscript should be updated to incorporate this data.
P3L64 - A recent single institution’s longitudinal analysis of UM patients with liver metastasis shows, however, a slight improvement in therapy outcome likely due to the introduction of liver directed therapies instead of dacarbazine based chemotherapy.
Citing a single centre retrospective review as representative of liver directed therapy is not be sufficient. Alternatively, any details are better reserved for part 4.
MOLECULAR CHARACTERISTICS
This section is comprehensive and full of interesting, well researched information. However, the flow needs to be improved as it feels like disconnected facts. Make sure the relevance of each sentence to uveal melanoma metastases is clear.
P3L87 - ..but apparently not via YAP/TAZ which, however, can determine resistance to BRAF- mutations have also been described for CM as a mechanism of resistance to BRAF inhibitors
Use of qualifiers, like ‘apparently’ need to be reviewed throughout the manuscript. Shoehorning of a acquired resistance mechanism is clumsy.
P3L102 - … is needed in addition to DNA copy…
“is needed” – these genomic events convey a greatly increased risk of subsequent metastases, but UM without these features do have a risk of metastases.
P4L110 - Chr8q harbors the oncogene MYC that is, however, not over-expressed in UM with chr8q gain
Rephrase.
P4P115 - Chr6p harbors the cluster of HLA genes
Interpret the impact/consequences for UM
PATHOPHYSIOLOGY
The opening paragraph of this section is an example of sentences that read like a list of facts rather than a coherent paragraph.
P5L159 – The two paragraphs on hepatic tropism make use of three references and can be better substantiated.
P5L188 - For immune-checkpoint inhibition, liver as a metastatic site is a poor prognostic factor across cancer types. Look at the papers on immune cells and liver metastases (e.g Lee Sci Immunology 2020).
P6L249 – For these single arm retrospective studies of anti-CTLA-4 and anti-PD-1 monotherapy there response rates are important (~5%) as survival is easily confounded and the relative merits of these treatments are hard to interpret as presented.
As mentioned above, more data is now available for tebentafusp. Also combination immune therapy is an alternative standard of care (Pelseter JCO 2020, Piulats JCO 2021) and deserves more line space.
Author Response
We thank the referees for their comments that have helped to greatly improve the manuscript. Please find our answers to the issues raised in point by point manner within the text here below and in the manuscript with revision marking.
Referee #1
The authors have done a commendable job with an extensive review of uveal melanoma metastases. It will make a valuable addition to the literature, but I believe it requires some additional work to improve its readability.
My global view of the manuscript is that the facts are not well linked together to tell a coherent narrative. There are also many occasions were language use is not precise enough – either using qualifiers (might, may, apparently) that don’t adequately reflect the data, or being more definitive than the evidence might support (e.g the premise about why therapeutic modalities do not work in uveal melanoma, these are certainly contributing factors but are not the whole story.
We have introduced many changes in the manuscript in order to make the narrative more coherent. We have revised qualifiers eliminating, where appropriate, those introducing uncertainty and weakening overstatements.
Specific, but not exhaustive, feedback follows:
INTRODUCTION
P3L59 - …that allow for a precise prognostication, unmatched by other cancers.
Provide additional support – is it truly unmatched?
We have cancelled this statement since we are not aware of systematic comparative studies that could justify this assertion.
P3L62 - there are FDA approved therapies neither for metastatic UM nor for adjuvant therapy
The phase III trial of tebentafusp was recently published (NEJM 9/2021) and has demonstrated an overall survival benefit. This technically has FDA accelerated approval. The manuscript should be updated to incorporate this data.
We have changed the paragraph in order to also incorporate the BLA of Tebentafusp granted by FDA.
P3L64 - A recent single institution’s longitudinal analysis of UM patients with liver metastasis shows, however, a slight improvement in therapy outcome likely due to the introduction of liver directed therapies instead of dacarbazine based chemotherapy.
Citing a single centre retrospective review as representative of liver directed therapy is not be sufficient. Alternatively, any details are better reserved for part 4.
We have moved this sentence to the discussion of liver directed therapy in chapter 4 where this is more extensively discussed.
MOLECULAR CHARACTERISTICS
This section is comprehensive and full of interesting, well researched information. However, the flow needs to be improved as it feels like disconnected facts. Make sure the relevance of each sentence to uveal melanoma metastases is clear.
The first paragraph of chapter 2 is meant as a very condensed introduction to the molecular characteristics of UM that are different from CM. We introduced the half sentence “…and these differences are believed to determine the drastically different responses to targeted and immune therapy” in order to indicate why this description is relevant to UM metastasis.
P3L87 - ..but apparently not via YAP/TAZ which, however, can determine resistance to BRAF- mutations have also been described for CM as a mechanism of resistance to BRAF inhibitors
Use of qualifiers, like ‘apparently’ need to be reviewed throughout the manuscript. Shoehorning of a acquired resistance mechanism is clumsy.
We have eliminated the references to BRAFi resistance that admittedly is not relevant in this context.
P3L102 - … is needed in addition to DNA copy…
“is needed” – these genomic events convey a greatly increased risk of subsequent metastases, but UM without these features do have a risk of metastases.
We changed this phrase accordingly.
P4L110 - Chr8q harbors the oncogene MYC that is, however, not over-expressed in UM with chr8q gain
Rephrase.
We have changed this sentence.
P4P115 - Chr6p harbors the cluster of HLA genes
Interpret the impact/consequences for UM
We have added considerations on the role of HLA in UM.
PATHOPHYSIOLOGY
The opening paragraph of this section is an example of sentences that read like a list of facts rather than a coherent paragraph.
We have thoroughly edited this paragraph in order to deliver a convincing narrative.
P5L159 – The two paragraphs on hepatic tropism make use of three references and can be better substantiated.
We have revised these paragraphs adding more information and references.
P5L188 - For immune-checkpoint inhibition, liver as a metastatic site is a poor prognostic factor across cancer types. Look at the papers on immune cells and liver metastases (e.g Lee Sci Immunology 2020).
We have added a related comment and the reference.
P6L249 – For these single arm retrospective studies of anti-CTLA-4 and anti-PD-1 monotherapy there response rates are important (~5%) as survival is easily confounded and the relative merits of these treatments are hard to interpret as presented.
We have modified these statements.
As mentioned above, more data is now available for tebentafusp. Also combination immune therapy is an alternative standard of care (Pelseter JCO 2020, Piulats JCO 2021) and deserves more line space.
We have up-dated the discussion of Tebentafuso and added the combination therapy data.
Reviewer 2 Report
In this review manuscript, the authors updated information on uveal melanoma metastasis from aspects of molecular characteristics, pathophysiology, therapy, models and outlook. Overall the review is interesting and will be a good reference for the researchers.
There are several aspects that may need to improve:
1) In the introduction the authors should give reasons why the focus of metastasis is chosen for this review. Information regarding the average time to develop metastasis, percentage of patients developing metastasis, etc, and a general overview of what is known about UM metastasis.
minor issue: line 59-61, unclear meaning. Enough molecular signature or not enough?
2) Metastasis-associated molecular characteristics of UM: this section seems to emphasize molecular characteristics of UM, instead of UM metastasis. The authors may need to compare primary tumor vs. metastatic tumor and tell the readers in fact what kind of molecular characteristics are associated with metastasis in UM. The current writing does not give this information.
3) Line 185: give brief description of immune privilege of eye, and how it allows UM cells to escape the immune surveillance.
4) In section 4 - Therapy of UM metastases: OS means median OS? same for PFS? After certain treatment, when the authors give OS, such as 4.6 months, please give the comparison group or give comments to state whether it is better than standard treatment or worse. A pile of data without adequate comments can be difficult to comprehend.
5) Line 289-292: using murine cutaneous melanoma cell lines to generate
liver metastasis? That sounds odd and inadequate.
6) Line 321, specify the two pathways.
7) the author compared CM and UM, and BAP1 mutation --here is a missing reference: J Clin Med. 2020 Feb; 9(2): 411.
Author Response
We thank the referees for their comments that have helped to greatly improve the manuscript. Please find our answers to the issues raised in point by point manner within the text here below and in the manuscript with revision marking.
Referee #2
In this review manuscript, the authors updated information on uveal melanoma metastasis from aspects of molecular characteristics, pathophysiology, therapy, models and outlook. Overall the review is interesting and will be a good reference for the researchers.
There are several aspects that may need to improve:
1) In the introduction the authors should give reasons why the focus of metastasis is chosen for this review. Information regarding the average time to develop metastasis, percentage of patients developing metastasis, etc, and a general overview of what is known about UM metastasis.
We added a paragraph to the introduction to better introduce the subject.
minor issue: line 59-61, unclear meaning. Enough molecular signature or not enough?
We have changed the sentence for major clarity.
2) Metastasis-associated molecular characteristics of UM: this section seems to emphasize molecular characteristics of UM, instead of UM metastasis. The authors may need to compare primary tumor vs. metastatic tumor and tell the readers in fact what kind of molecular characteristics are associated with metastasis in UM. The current writing does not give this information.
We have added the missing referral to studies on UM metastases and we have modified the text in order to make the effect of somatic mutations on UM progression more evident.
3) Line 185: give brief description of immune privilege of eye, and how it allows UM cells to escape the immune surveillance.
We have added this information.
4) In section 4 - Therapy of UM metastases: OS means median OS? same for PFS? After certain treatment, when the authors give OS, such as 4.6 months, please give the comparison group or give comments to state whether it is better than standard treatment or worse. A pile of data without adequate comments can be difficult to comprehend.
We specified that OS is median OS and PFS is median PFS. We also specified that for metastatic uveal melanoma none of the treatment can yet be considered standard.
5) Line 289-292: using murine cutaneous melanoma cell lines to generate
liver metastasis? That sounds odd and inadequate.
We agree, yet this has been initially be done given the lack of more appropriate models.
6) Line 321, specify the two pathways.
We have specified MAP-kinase and YAP/TAZ pathways.
7) the author compared CM and UM, and BAP1 mutation --here is a missing reference: J Clin Med. 2020 Feb; 9(2): 411.
We have added this reference.
Reviewer 3 Report
Dear authors,
thank you for this excellent review on uveal melanoma. To add to the picture I would suggest to also mention the successful treatment of uveal melanoma with dendritic cell vaccination and the ongoing prospective clinical trial investigating the efficacy of DC-vaccination.
Furthermore I would suggest to use another expression in line 61/62: it is not entirely clear what "keeping the path" means in the context.
Kind regards
Author Response
We thank the referees for their comments that have helped to greatly improve the manuscript. Please find our answers to the issues raised in point by point manner within the text here below and in the manuscript with revision marking.
Referee #3
thank you for this excellent review on uveal melanoma. To add to the picture I would suggest to also mention the successful treatment of uveal melanoma with dendritic cell vaccination and the ongoing prospective clinical trial investigating the efficacy of DC-vaccination.
The treatment with DC-vaccination and the ongoing trial have been mentioned.
Furthermore I would suggest to use another expression in line 61/62: it is not entirely clear what "keeping the path" means in the context.
We have changed the sentence in order to make it clearer.
Round 2
Reviewer 1 Report
I would like to thank the authors for the time taken to revise this manuscript. The factual contents are generally acceptable.
However the manuscript still requires careful editing and review regarding use of english to improve readability.
There has been a substantial amount of work put into this manuscript, and to be fair, I have reviewed previous papers from the first and corresponding authors and these were to higher written standard so I know this can be met.
Author Response
We thank the Referee for their positive consideration of our work and for their helpful hints. We have done our best to improve the readibility and we have carefully checked for errors.
As asked by the Editor, we have also added a figure on molecular features of high-risk primitive uveal melanoma and a table on the therapies tested for metastatic UM.